# Dithioethanol (DTE)-Conjugated Deoxyribose Cyclic Dinucleotide Prodrugs (DTE-dCDNs) as STING Agonist

**DOI:** 10.3390/ijms25010086

**Published:** 2023-12-20

**Authors:** Zhiqiang Xie, Yuchen Yang, Zhenghua Wang, Dejun Ma, Zhen Xi

**Affiliations:** 1State Key Laboratory of Elemento-Organic Chemistry, Department of Chemical Biology, College of Chemistry, Nankai University, Tianjin 300071, China; 1120200432@mail.nankai.edu.cn (Z.X.); 2120200933@mail.nankai.edu.cn (Y.Y.); wzhnren@163.com (Z.W.); madejun@nankai.edu.cn (D.M.); 2Frontiers Science Center for New Organic Matter, Nankai University, Tianjin 300071, China

**Keywords:** DTE-conjugated prodrug, deoxyribose cyclic dinucleotide, STING agonist, intracellular signaling, cancer immunotherapy

## Abstract

To improve the chemical regulation on the activity of cyclic dinucleotides (CDNs), we here designed a reduction-responsive dithioethanol (DTE)-based dCDN prodrug **9** (DTE-dCDN). Prodrug **9** improved the cell permeability with the intracellular levels peaking in 2 h in THP-1 cells. Under the reductive substance such as GSH or DTT, prodrug **9** could be quickly decomposed in 30 min to release the parent dCDN. In THP1-Lucia cells, prodrug **9** also retained a high bioactivity with the EC_50_ of 0.96 μM, which was 51-, 43-, and 3-fold more than the 2′,3′-cGAMP (EC_50_ = 48.6 μM), the parent compound 3′,3′-c-di-dAMP (EC_50_ = 41.3 μM), and ADU-S100 (EC_50_ = 2.9 μM). The high bioactivity of prodrug **9** was validated to be highly correlated with the activation of the STING signaling pathway. Furthermore, prodrug **9** could also improve the transcriptional expression levels of *IFN-β*, *CXCL10*, *IL-6,* and *TNF-α* in THP-1 cells. These results will be helpful to the development of chemically controllable CDN prodrugs with a high cellular permeability and potency.

## 1. Introduction

The cyclic GMP-AMP synthase (cGAS)–STING pathway has emerged as an important intrinsic tumor-sensing mechanism [1,2]. Tumor-derived DNA activates cGAS to produce 2′,3′-cGAMP, the endogenous ligand of STING, resulting in the downstream signaling cascade via the recruitment of threonine-protein kinase (TBK1), phosphorylation of the interferon regulatory transcription factor IRF3, and production of type I interferon (IFN), and other proinflammatory cytokines. In parallel, STING and TBK1 also interact with the inhibitors of κB kinases (IKKs), and then release the transcription factor NF-κB from its inhibitor, thus allowing the transcription of proinflammatory cytokines and chemokines such as tumor necrosis factor-α (TNFα) and interleukin 6 (IL6), via the nuclear factor κ-light-chain-enhancer of the activated B cells (NF-κB) pathway [3,4]. These events can selectively stimulate the cross-presentation of tumor antigens and mobilization of tumor-specific CD8^+^ T cells, which prime the adaptive immune response against tumors [5,6]. Accordingly, STING has been widely investigated as a therapeutic target for the treatment of infectious diseases, cancer immunotherapy, and autoimmunity, as well as vaccine adjuvants [7,8,9,10].

Owing to some limitations of endogenous cyclic dinucleotides (CDNs), such as rapid clearance and poor membrane permeability, there has been an increased interest in identifying new STING agonists with improved drug-like properties compared to natural STING ligands [11,12,13]. However, the cell delivery and bioactivity of CDNs were still unmet in clinical trials, which need further structure optimization.

The finding and design of different structures of CDNs were the main goals to enhance bioactivity, selectivity, stability, and safety [14,15,16]. In general, CDNs are polar molecules bearing two negative charges on the phosphoric oxygen atoms, and their cellular uptake requires the presence of transporters since the free diffusion of polar molecules via the lipid cell membrane is limited [17]. For a successful drug design, the negative charges of CDNs need to be masked to increase their ability to enter cells via diffusion through the cell membrane. Prodrug strategies were developed to improve the transport of polar nucleoside phosphates and phosphonates across the cell membrane to cells, where the biolabile protecting groups are enzymatically or chemically cleaved in an intracellular environment and the active parent species is released [18,19,20,21].

In our previous work, we have designed esterase-responsive SATE-dCDN prodrugs with a high cell uptake efficiency and high bioactivity [21]. Since the bioactivity of SATE-dCDN strictly relies on the abundance of cellular esterase, the programable regulation of their activity was mostly limited in multiple tissues, which will add the risk of systemic toxicity. To further alleviate the potential risk, a naturally regulated and selective release strategy could be introduced. Reduced glutathione (GSH) is an important substance that determines the redox environment in organisms. It is worth noting that the concentration of GSH in tumor tissues was at least four times higher than that in normal tissues, and the intracellular concentration of GSH was up to 2–10 mmol/L, which is an ideal trigger for drug delivery [22].

Considering that disulfide bond is one of the most widely used redox-responsive connecting arms [23,24,25], in this study, we designed the dithioethanol (DTE)-based dCDN prodrug **9** (DTE-dCDN) to test cell uptake, controllable release, cellular bioactivity, and toxicity. For prodrug **9**, when the DTE group masked the negatively charged phosphodiester group of dCDN, it helped CDN cross the cell membrane easily. Once these prodrugs enter the cell, the DTE group was then degraded, which was initiated by reductive reagents. As the disulfide bridge broke, the unstable intermediate with *O*-2-mercaptoetnylphosphotriester would decompose spontaneously via intermolecular nucleophilic displacement into the corresponding phosphodiester structure dCDN (Figure 1). Immediately after that, the free dCDN would stimulate the cGAS-STING pathway to induce the production of type I IFNs and proinflammatory cytokines. The design and bioactivity evaluation of **9** will provide an effective approach to chemically regulate the activity of CDNs.

## 2. Results and Discussion

### 2.1. Synthesis of Prodrug ***9***

Prodrug **9** was synthesized, as shown in Figure 1. Compound **3** was obtained by coupling deoxynucleoside phosphoramidite **2** and 3-hydroxypropionitrile with the activator ETT in MeCN and then oxidized with TBHP. Only one cyanoethyl group for compound **3** was removed when treated with the *t*BuNH_2_-MeCN mixture, and then reacted with compound **1** with MSNT in anhydrous pyridine to form the intermediate **4**. The cyanoethyl group for compound **4** was removed under the t-BuNH_2_/MeCN mixture to achieve intermediate **5** without further purification. The removal of the 5′-DMTr protecting groups of compounds **4** with a solution of 6% dichloroacetic acid (DCA) in dichloromethane provided the intermediate **6**. Subsequently, a combination of **5** and **6** in the presence of coupling reagent MSNT in pyridine and treatment with 6% DCA solution could generate the corresponding linear dinucleotide **7**. After removing the cyanoethyl with the *t-*BuNH_2_/MeCN mixture, the cyclization was achieved with MSNT in pyridine to obtain the cyclic dinucleotide phosphotriester **8**, which was then treated with 10% diisopropylamine (DIA) in methanol to achieve the DTE-dCDN prodrug **9** in a 12% overall yield.

### 2.2. Fast Uptake of DTE-dCDN Prodrug into THP-1 Cells

To determine the cell uptake of the DTE-dCDN prodrug, prodrug **9** and its parent dCDN were prepared. Prodrug **9** entered THP-1 cells rapidly and was efficiently metabolized to its parent dCDN 3′,3′-c-di-dAMP as determined via liquid chromatograph MS/MS analysis. As shown in Figure 2, the intracellular levels of prodrug **9** peaked in 2 h but then rapidly decreased, the intracellular levels of the parent dCDN 3′,3′-c-di-dAMP peaked in 6 h, while the intracellular levels of the parent dCDN 3′,3′-c-di-dAMP were not detectable within 6 h when the THP-1 cells were treated with 3′,3′-c-di-dAMP even at a 10× higher concentration than was used in the case of prodrug **9**. Contrary to the rapid entry of the prodrug, the parent CDN was barely detectable in cells after a 6 h incubation, and its intracellular concentration increased linearly for up to 24 h. These results indicated that the DTE-dCDN prodrug improved the cell permeability and could be quickly decomposed to release the parent dCDN in cell cytoplasm.

### 2.3. DTE-dCDN Prodrug Showed Reduction-Dependent Release of Parent dCDN

To analyze the reduction dependency, we performed the in vitro cleavage test by incubating prodrug **9** with a reductive substance such as GSH or DTT and cell lysates at 37 °C and then determined the change with HPLC. As shown in Figure 3, the elution time of prodrug **9** and its parent dCDN 3′,3′-c-di-dAMP were 6.0 min and 1.2 min, respectively. As the reductive substance or cell lysates were added, the peak for prodrug **9** disappeared and the new peaks at 1.2 min appeared in 0.5 h and 2 h, respectively. The released parent dCDN was also confirmed with 3′,3′-c-di-dAMP through UPLC-MS (Appendix A). The results indicated that the DTE-dCDN prodrug **9** could be quickly decomposed to release the parent dCDN in a reductive substance such as DTT or GSH.

### 2.4. Parent dCDN of DTE-dCDN Prodrug ***9*** Exhibited the Improved Serum Stability

The serum stability of the DTE-dCDN prodrug **9**, parent dCDN 3′,3′-c-di-dAMP, 2′,3′-cGAMP, and ADU-S100 was tested in 20% fetal bovine serum (FBS). As shown in Figure 4 and Appendix A, the DTE-dCDN prodrug **9** could be quickly degraded into the parent dCDN in 20% FBS, while the parent dCDN 3′,3′-c-di-dAMP exhibited improved serum stability. The parent dCDN (3′,3′-c-di-dAMP) did not show any degradation at 72 h while the 80% 2′,3′-cGAMP and 66% ADU-S100 were degraded at 72 h, which indicated the backbone of the parent dCDN contributed to improving the stability.

### 2.5. DTE-dCDN Prodrug ***9*** Maintained High Bioactivity in THP1-Lucia Cells

For the bioactivity assay, we measured the luciferase activity induced by prodrug **9** in THP1-Lucia cells. The endogenous 2′,3′-cGAMP, the parent dCDN 3′,3′-c-di-dAMP, and ADU-S100 were set as the control group. As shown in Figure 5, the activity of prodrug **9** (EC_50_ = 0.96 μM ± 0.004 μM) was around 51-, 43-, and 3.0-fold more than the 2′,3′-cGAMP (EC_50_ = 48.6 ± 1.1 μM), 3′,3′-c-di-dAMP (EC_50_ = 41.3 ± 2.5 μM), and ADU-S100 (EC_50_ = 2.9 ± 0.2 μM). Furthermore, the maximum activation fold (E_max_) of **9** was approximately 1.8-, 1.9-, and 1.4-fold higher than the 2′,3′-cGAMP, 3′,3′-c-di-dAMP, and ADU-S100, respectively. The DTE-based prodrug improved the bioactivity of the parent dCDN and exhibited a higher bioactivity compared to the 2′,3′-cGAMP and ADU-S100 in the THP1-Lucia cells.

### 2.6. Prodrug ***9*** Relied on STING-Dependent Signal Transduction for the IFN-β Activation

To demonstrate the dependence of STING on the activity of prodrug **9**, we incubated THP1-Lucia cells with prodrug **9** alone or pretreated with the STING and TBK1 inhibitor to measure the luciferase activity. The endogenous 2′,3′-cGAMP was set as the positive control group. H-151 [26] and BX795 [27] were selected to use as the STING inhibitor and TBK1 inhibitor, respectively. As shown in Figure 6, the inhibition of the downstream effector STING and TBK1, with H-151 and BX795, could block the induction of the luciferase expression by prodrug **9**, which was in accordance with the test of the 2′,3′-cGAMP. The results indicated that prodrug **9** could stimulate the IFN-β promoter responsive luciferase expression through the STING signaling pathway.

### 2.7. Prodrug ***9*** Stimulated the Production of Type I Interferons and Proinflammatory Cytokines in THP-1 Cells

The THP-1 cells bearing both the ISG reporter and NF-κB reporter were used to determine the activation of prodrug **9** on the STING signaling. The activation of STING led to self-phosphorylation and started the downstream target gene transcription, which produced type I interferons and proinflammatory cytokines, such as CXCL10, IL6, and TNF-α [28,29]. To further validate the immunostimulatory activity of **9**, we detected the above four tumor-related cytokines to explore the effect of prodrug **9** on the STING signaling pathway. Real-time quantitative PCR analysis (RT-qPCR) was employed to examine the mRNA levels of *IFN-β*, *CXCL10*, *IL-6,* and *TNF-α* in the THP-1 cells treated with prodrug **9**, 3′,3′-c-di-dAMP, 2′,3′-cGAMP, and ADU-S100. As shown in Figure 7, prodrug **9** induced the expression of *IFN-β*, *CXCL10*, *IL-6,* and *TNF-α* by approximately 43-, 26-, 1.4-, and 12-fold compared to 3′,3′-c-di-dAMP, respectively. Compared to 2′,3′-cGAMP, prodrug **9** induced the expression of *IFN-β*, *CXCL10*, *IL-6,* and *TNF-α* by approximately 40-, 17-, 1.2-, and 46-fold, respectively. Meanwhile, the expression of *IFN-β*, *CXCL10*, *IL-6,* and *TNF-α* by prodrug **9** was also 14-, 14-, 1.9-, and 15-fold higher than that by ADU-S100, respectively. The results indicated that prodrug **9** could activate the STING signaling pathway to promote the expression levels of IFN-β, CXCL10, IL-6, and TNF-α in THP-1 cells.

In order to validate the STING activation in other cells like tumor cells, we chose colon cancer cells CT-26 as a model to investigate the effect of these compounds on the STING activation-mediated *IFN-β* gene expression. Colon cancer cells CT-26 showing the extremely low expression of STING proteins are usually used as a tumor model to verify the antitumor immunotherapeutic effect in mice. Under the treatment of prodrug **9** and CDNs, the mRNA expression levels of *IFN-β* were also assessed via real-time quantitative PCR (qPCR). As seen in Appendix A, all the compounds including ADU-S100 exhibited no significant induction on the transcription of *IFN-β* in CT-26 cells, which was contrary to the high bioactivity in THP-1 cells. It suggested that prodrug **9** exhibited its bioactivity relying on the normal STING expression as seen in THP-1 cells.

Next, we employed the ELISA method to examine the actual protein levels of IFN-β in the THP-1 cells treated with prodrug **9**, 3′,3′-c-di-dAMP, 2′,3′-cGAMP, and ADU-S100. As shown in Appendix A, prodrug **9** induced the protein levels of IFN-β by approximately 66-, 4.3-, and 5.6-fold compared to 2′,3′-cGAMP, 3′,3′-c-di-dAMP, and ADU-S100. The results of the ELISA are consistent with those from the qPCR analysis. Hence, prodrug **9** could induce the STING signaling pathway to increase the transcriptional and protein expression levels of IFN-β and other proinflammatory cytokines.

### 2.8. Cell Viability of Prodrug ***9*** in THP-1 Cells and HEK293T Cells

To evaluate the safety of the designed prodrug, cell counting kit-8 (CCK8) was used to quantitatively assess the cell viability of the DET-dCDN prodrug **9**, 3′,3′-c-di-dAMP, and 2′,3′-cGAMP. The CCK8 assay revealed that prodrug **9** has no significant influence on the growth of THP-1 and HEK293T cells (Figure 8). It suggested that prodrug **9** showed an undetectable cytotoxicity to THP-1 cells and HEK293T cells.

### 2.9. Western Blot Analysis of STING Downstream Signaling Pathways in THP-1 Cells

To determine the activation of the STING signaling by prodrug **9**, we analyzed its effect on the phosphorylation of the downstream STING/TBK1/IRF3 signaling pathway. The THP-1 cells were treated with prodrug **9**, 2′,3′-cGAMP, 3′,3′-c-di-dAMP, and ADU-S100 for 4 h, and the levels of the total STING, phospho-STING, total TBK1, phospho-TBK1, total IRF3, phospho-IRF3, and β-actin were assessed via Western blotting. As shown in Figure 9 and Appendix A, prodrug **9** induced the relative expression of phospho-STING, phospho-TBK1, and phospho-IRF3 by 20.8-, 6.9-, and 19.8-fold compared to 2′,3′-cGAMP, respectively. Compared to 3′,3′-c-di-dAMP, prodrug **9** also induced the relative expression of phospho-STING, phospho-TBK1 and phospho-IRF3 by 5.5-, 3.6-, and 6.5-fold, respectively. Meanwhile, compared to ADU-S100, prodrug **9** induced the relative expression of phospho-STING, phospho-TBK1, and phospho-IRF3 by 1.6-, 1.3-, and 1.8-fold, respectively. The results showed that prodrug **9** dramatically increased the phosphorylation of STING, TBK1, and IRF3 after 4 h of treatment, indicating the capacity to stimulate the STING signaling pathway of prodrug **9**.

## 3. Materials and Methods

### 3.1. Chemistry

Unless otherwise specified, all solvents and reagents were purchased from commercial sources and used without further purification. Reactions were monitored by TLC on silica gel GF254 with detection under UV light. Column chromatography was performed using 300–400 mesh or 200–300 mesh silica gel. NMR spectra were recorded on Bruker AVANCE 400 M instrument (Chemical characterization of ^1^H, ^13^C, ^31^P NMR and HPLC spectra can be seen in Appendix A). Chemical shifts (δ) were reported in ppm downfield from an internal TMS standard, and J values were given in Hz. The following abbreviations were used to explain the multiplicities: s (singlet), d (doublet), t (triplet), q (quartet), m (multiplet). The number of protons (n) for a given resonance were indicated as nH. HRMS (MALTI-TOF) was obtained from Varian 7.0T FTMS. HPLC was achieved using Agilent 1260 (Agilent Technologies, Santa Clara, CA, USA). Preparative HPLC was performed on an Agela OCTOPUS purification system with monitoring at 254 nm on an ASB C18 column (10 μm OBD, 21.2 × 250 mm) (Agela Technologies, Tianjin, China) using gradients of H_2_O and MeCN at a flow rate of 10 mL/min. Purity of all final compounds tested in biological assays was determined to be >95% by HPLC analysis. The following analytical method was used to determine the chemical purity of the final compounds: HPLC, Agilent 1260, 10 mM TEAA buffer (mobile phase A), MeCN (mobile phase B), column Agilent ZORABX SB-C18 5 μm [4.6 × 150 mm], column temperature 25 °C, 1 mL/min, 254 nm. The gradient was as follows: 0–2 min: 98% A/2% B; 2–6 min: 98% A/2% B to 100% B; 6–10 min: 100% B; 10–13 min: 100% B to 98% A/2% B; 13–15 min: 98% A/2% B. 3′,3′-c-di-AMP and 2′,3′-cGAMP were prepared according to the literature procedures [15]. ADU-S100 was purchased from MedChemExpress (MCE, Monmouth Junction, NJ, USA, Cat. No: HY-12885B).

Synthesis of Compound **3**. Deoxynucleoside phosphoramidite (**2**) (0.99 g, 1.12 mmol), 3-Hydroxypropionitrile (85 mg, 1.20 mmol), and ETT (440 mg, 3.38 mmol) were dissolved in dry MeCN (20 mL). The mixture was stirred at room temperature for 1 h under Ar atmosphere. An amount of 1 mL TBHP (5.5 M in decane) was added for another 40 min. Afterwards, the solvent was removed under reduced pressure and the residue was resolved with 20 mL DCM. The reaction was washed with a saturated aqueous solution of NaHCO_3_ and extracted with DCM (2 × 20 mL). The organic phase was combined, washed with NaCl saturated solution, dried over anhydrous Na_2_SO_4_, filtered, and concentrated. The residue was purified via chromatography on silica gel (DCM: MeOH = 100:1~50:1) to give compound **3** as a white solid (890 mg, yield 91.0%).

**3**: ^1^H NMR (400 MHz, CDCl_3_) δ 9.54 (s, 1H), 8.71–8.65 (m, 1H), 8.21–8.11 (m, 1H), 7.41–7.19 (m, 12H), 7.04 (dd, *J* = 7.6, 3.6 Hz, 3H), 6.79 (td, *J* = 8.8, 1.6 Hz, 4H), 6.49 (t, *J* = 6.8 Hz, 1H), 5.34–5.25 (m, 1H), 4.87 (s, 2H), 4.46–4.19 (m, 4H), 3.76 (d, *J* = 6.6 Hz, 6H), 3.51–3.38 (m, 2H), 3.16–3.30 (m, 1H), 2.86–2.67 (m, 4H), 2.53–2.59 (m, 1H). ^13^C NMR (101 MHz, CDCl_3_) δ 166.90, 158.66, 158.56, 157.07, 152.45, 151.45, 151.42, 148.41, 148.31, 144.46, 144.29, 142.16, 142.07, 135.64, 135.38, 135.30, 130.08, 130.04, 129.85, 128.11, 127.96, 127.90, 127.10, 126.95, 123.15, 122.42, 118.31, 116.42, 116.37, 114.96, 113.25, 113.18, 86.84, 86.55, 86.45, 86.42, 84.93, 84.87, 84.74, 84.70, 79.73, 79.68, 72.31, 68.17, 63.73, 63.03, 62.71, 62.66, 62.60, 57.74, 55.28, 55.24, 53.85, 40.19, 38.02, 37.98, 21.52, 19.75, 19.70, 19.67, 19.63. ^31^P NMR (162 MHz, CDCl_3_) δ −3.04, −3.21. MALDI-TOF-HRMS: calcd for C_45_H_44_N_7_NaO_10_P [M + Na]^+^ 896.2779, found: 896.2780.

Synthesis of Compound **4**. In a 50 mL round flask, compound 3 (1.20 g, 1.37 mmol) was dissolved in 10 mL MeCN and 3 mL tert-butylamine and then stirred at room temperature for 20 min. The solvent was evaporated under reduced pressure. Compound 1 (260 mg, 1.37 mmol) and MSNT (1.22 g, 4.12 mmol) were added into the flask, the mixture was dissolved in 10 mL anhydrous pyridine, and stirred at room temperature under Ar atmosphere overnight. Several drops of water were added into the flask to stop the reaction. Afterwards, the solvent was removed under reduced pressure and resolved with 20 mL DCM again. Oxalic acid aqueous solution was added into the mixture. The organic phase was separated, and the water phase was extracted with DCM (2 × 50 mL). The organic phase was combined and washed with water twice (2 × 50 mL), dried over anhydrous Na_2_SO_4_, filtered, and concentrated. The residue was purified via chromatography on silica gel (DCM: MeOH = 100:1~50:1) to give compound 4 as a white solid (946 mg, yield 69.8%).

**4**: ^1^H NMR (400 MHz, CDCl_3_) δ 9.72–9.43 (m, 1H), 8.78 (s, 1H), 8.69 (s, 1H), 8.52–8.44 (m, 1H), 8.18 (dd, *J* = 13.4, 3.1 Hz, 1H), 8.04 (s, 1H), 7.71–7.61 (m, 2H), 7.42–7.33 (m, 4H), 7.32–7.23 (m, 6H), 7.21–7.17 (m, 2H), 7.11–7.02 (m, 4H), 6.92–6.76 (m, 4H), 6.54–6.40 (m, 1H), 5.41–5.26 (m, 1H), 4.89 (d, *J* = 5.5 Hz, 2H), 4.54–4.18 (m, 5H), 3.99 (d, *J* = 12.9 Hz, 1H), 3.88 (d, *J* = 12.9 Hz, 1H), 3.84 (s, 1H), 3.82–3.76 (m, 5H), 3.46 (dd, *J* = 17.1, 3.0 Hz, 1H), 3.25–3.01 (m, 3H), 2.88–2.63 (m, 3H). ^13^C NMR (101 MHz, CDCl_3_) δ 159.66, 158.82, 158.79, 158.59, 158.55, 156.94, 152.03, 150.57, 149.80, 149.09, 147.36, 146.87, 144.23, 143.14, 141.13, 139.48, 137.32, 135.27, 132.51, 131.39, 130.03, 130.00, 129.85, 129.52, 129.13, 128.70, 128.32, 128.07, 127.92, 127.81, 127.76, 127.03, 124.06, 122.46, 121.26, 121.16, 120.23, 116.47, 116.43, 114.92, 114.90, 113.55, 113.20, 113.13, 113.12, 113.10, 113.08, 113.00, 87.75, 87.69, 87.28, 81.36, 80.69, 80.09, 68.09, 65.92, 65.87, 65.83, 62.99, 62.45, 62.40, 55.36, 55.24, 39.07, 38.23, 38.16, 19.84, 19.77. ^31^P NMR (162 MHz, CDCl_3_) δ −2.64, −2.87, −2.90, −3.22. MALDI-TOF-HRMS: calcd for C_49_H_48_N_7_NaO_10_PS_2_ [M + Na]^+^ 1012.2534, found: 1012.2540.

Synthesis of Compound **6**. In a 50 mL round flask, compound **4** (500 mg, 0.51 mmol) was dissolved in 10 mL DCM and stirred under an ice-water bath. An amount of 10 mL solution of 6% dichloroaceticacid in DCM was added into the mixture and continuously stirred for 5 min. Afterwards, several drops of MeOH were added. Saturated solution of sodium bicarbonate was added to regulate the pH to neutral. The organic phase was separated, and the water phase was extracted with DCM (2 × 20 mL). The organic phase was combined and washed with sodium bicarbonate saturated solution and water, then dried over anhydrous sodium sulfate, filtered, and concentrated. The residue was purified via chromatography on silica gel (DCM: MeOH = 100:1~30:1) to give compound **6** as a white solid (306 mg, yield 87.3%).

**6**: ^1^H NMR (400 MHz, CDCl_3_) δ 9.92 (s, 1H), 8.72 (d, *J* = 3.8 Hz, 1H), 8.54–8.41 (m, 1H), 8.35 (d, *J* = 3.6 Hz, 1H), 7.75–7.57 (m, 2H), 7.31 (dd, *J* = 12.8, 4.4 Hz, 3H), 7.10–7.18 (m, 1H), 7.01 (t, *J* = 7.6 Hz, 3H), 6.46 (dd, *J* = 9.2, 5.5 Hz, 1H), 5.36 (t, *J* = 5.6 Hz, 1H), 4.96 (s, 2H), 4.57–4.20 (m, 6H), 3.78 –4.05 (m, 2H), 3.12 (t, *J* = 6.2 Hz, 3H), 2.82 (t, *J* = 6.0 Hz, 2H), 2.71 (dd, *J* = 14.1, 5.1 Hz, 1H). ^13^C NMR (101 MHz, CDCl_3_) δ 167.57, 158.79, 158.77, 157.08, 151.83, 150.62, 149.76, 148.95, 143.37, 143.34, 137.28, 137.25, 129.74, 123.60, 122.20, 121.21, 120.09, 116.69, 116.65, 114.80, 114.76, 87.48, 87.44, 86.81, 86.78, 80.44, 80.39, 80.34, 68.24, 65.87, 65.84, 65.81, 65.78, 62.76, 62.51, 62.49, 62.46, 62.44, 53.58, 39.10, 39.06, 38.20, 38.12, 19.78, 19.72. ^31^P NMR (162 MHz, CDCl_3_) δ −2.72, −2.93, −2.98, −3.26. MALDI-TOF-HRMS: calcd for C_28_H_30_N_7_NaO_8_PS_2_ [M + Na]^+^ 710.1227, found: 710.1230.

Synthesis of linear dinucleotide **7**. In a 50 mL flask, compound **3** (297 mg, 0.30 mmol) was dissolved in 6 mL MeCN and 2 mL tert-butylamine and then stirred at room temperature for 20 min. The solvent was evaporated under reduced pressure. Compound **6** (206 mg, 0.30 mmol) and MSNT (448 mg, 1.51 mmol) were added into the flask, the mixture was dissolved in 10 mL anhydrous pyridine, and stirred at room temperature under Ar atmosphere overnight. Several drops of water were added into the flask to stop the reaction. Afterwards, the solvent was removed under reduced pressure and resolved with 20 mL DCM again. Oxalic acid aqueous solution was added into the mixture. The organic phase was separated, and the water phase was extracted with DCM (2 × 20 mL), followed by the combination of the organic phase, washing with water twice (2 × 20 mL), drying over anhydrous Na_2_SO_4,_ and filtration. The solvent was removed under reduced pressure and the residue was redissolved with 15 mL DCM and stirred under an ice-water bath. A solution of 6% dichloroaceticacid in DCM (15 mL) was added into the mixture and continuously stirred for 5 min. Afterwards, several drops of MeOH were added. Saturated solution of sodium bicarbonate was added to regulate the pH to neutral. The organic phase was separated, and the water phase was extracted with DCM (2 × 20 mL). The organic phase was combined, washed with sodium bicarbonate saturated solution and water, dried over anhydrous sodium sulfate, filtered, and concentrated. The residue was purified via chromatography on silica gel (DCM: MeOH = 100:1~20:1) to give compound **7** as a white solid (210 mg, yield 53.5%).

**7**: ^1^H NMR (400 MHz, CDCl_3_) δ 9.61 (s, 1H), 8.79 (d, *J* = 5.9 Hz, 1H), 8.73 (d, *J* = 5.1 Hz, 1H), 8.52–8.41 (m, 2H), 8.39–8.31 (m, 2H), 7.72–7.59 (m, 4H), 7.37–7.29 (m, 3H), 7.16–6.93 (m, 8H), 6.61–6.52 (m, 1H), 6.51–6.39 (m, 1H), 5.42 (s, 1H), 5.35–5.25 (m, 2H), 4.88 (s, 4H), 4.54 (s, 1H), 4.50–4.24 (m, 9H), 3.98–3.76 (m, 2H), 3.25–3.01 (m, 6H), 2.91–2.77 (m, 3H), 2.65 (dd, *J* = 13.6, 5.2 Hz, 1H). ^13^C NMR (101 MHz, CDCl_3_) δ 166.90, 158.84, 158.77, 158.68, 156.99, 156.93, 152.65, 151.97, 151.45, 150.60, 149.81, 149.76, 148.95, 148.53, 148.49, 143.50, 143.46, 142.16, 137.34, 129.86, 129.82, 129.80, 122.43, 122.38, 121.32, 121.23, 120.30, 120.29, 120.19, 116.64, 116.61, 114.91, 114.86, 114.82, 87.54, 87.05, 84.60, 83.73, 80.42, 78.22, 68.13, 66.07, 66.01, 65.86, 65.81, 62.87, 62.66, 62.61, 53.49, 38.99, 38.20, 38.13, 29.70, 19.89, 19.82. ^31^P NMR (162 MHz, CDCl_3_) δ −2.29, −2.33, −2.49, −2.70, −2.76. MALDI-TOF-HRMS: calcd for C_53_H_55_N_13_NaO_15_P_2_S_4_ [M + Na]^+^ 1326.2196, found: 1326.2195.

Synthesis of cyclic dinucleotide phosphotriester **8**. In a 50 mL flask, compound **7** (300 mg, 0.23 mmol) was dissolved in 6 mL MeCN and 2 mL tert-butylamine and then stirred at room temperature for 20 min. The solvent was evaporated under reduced pressure. An amount of 10 mL anhydrous pyridine was added after the residual was fully dried. MSNT (409 mg. 1.38 mmol) was added in batches. The mixture was stirred at room temperature overnight. The reaction was stopped with the addition of drops of water. The solvent was removed under reduced pressure and the residual was dissolved in 30 mL DCM, washed with oxalic acid aqueous solution, dried over anhydrous sodium sulfate, filtered, and concentrated. Further purification with chromatography on silica gel (DCM: MeOH = 100:1~15:1) gave the fully protected cyclic dinucleotide **8** as a white solid (180 mg, yield 63.4%).

**8**: ^1^H NMR (400 MHz, CDCl_3_) δ 9.76 (s, 2H), 8.76 (d, *J* = 9.7 Hz, 2H), 8.53–8.36 (m, 2H), 8.36–8.18 (m, 2H), 7.68–7.55 (m, 3H), 7.37–7.26 (m, 5H), 7.14–6.98 (m, 8H), 6.54–6.40 (m, 2H), 5.71–5.36 (m, 2H), 4.93 (s, 4H), 4.66–4.32 (m, 9H), 4.12 (d, *J* = 5.1 Hz, 1H), 3.67–3.34 (m, 2H), 3.11 (t, *J* = 6.2 Hz, 4H), 2.82–2.61 (m, 2H), 1.26 (s, 2H). ^13^C NMR (101 MHz, CDCl_3_) δ 167.30, 158.86, 158.68, 158.57, 157.10, 157.06, 152.49, 152.42, 152.35, 151.38, 151.33, 151.24, 151.16, 149.81, 148.67, 148.60, 143.12, 142.92, 142.76, 142.55, 137.22, 137.18, 129.77, 123.49, 123.39, 123.28, 122.28, 122.26, 121.19, 120.10, 120.08, 114.85, 114.55, 85.58, 85.23, 83.03, 82.59, 79.03, 78.17, 77.84, 68.21, 66.14, 66.09, 65.93, 65.88, 65.44, 64.99, 53.52, 38.29, 38.23, 38.12, 38.05, 38.00, 36.29, 31.88, 29.65, 29.62, 29.32, 22.66, 14.13. ^31^P NMR (162 MHz, CDCl_3_) δ −1.00, −1.29, −4.32. MALDI-TOF-HRMS: calcd for C_50_H_50_N_12_NaO_14_P_2_S_4_ [M + Na]^+^ 1255.1825, found: 1255.1825.

Synthesis of prodrug **9**. Compound **7** (99 mg, 0.08 mmol) was dissolved in 5 mL MeOH in a 20 mL flask. An amount of 500 μL of DIA was added and the mixture was stirred for 3 h. The solvent was removed under reduced pressure. The crude was purified with the preparative reverse-phase HPLC [Agela OCTOPUS purification System(Agela & Phenomenex, Tianjin, China); Preparative column using an ASB C18 column (21.2 (diameter) mm × 250 (height) mm) (Agela Technologies); A = water, B = MeCN; gradient: 0–2 min: 60% A/40% B; 2–32 min: 60%A to 0% A/40% B to 100% B; 32–37 min: 100% B] to give prodrug **9** as a white solid (50 mg, yield 63.9%).

**9**: HPLC: t_R_ = 6.02 min, purity: 96.5%. ^1^H NMR (400 MHz, DMSO) δ 8.48–8.34 (m, 3H), 8.16 (d, *J* = 2.4 Hz, 1H), 7.84–7.73 (m, 3H), 7.35 (s, 3H), 7.27–7.16 (m, 2H), 6.53–6.30 (m, 2H), 5.48 (s, 1H), 5.34 (s, 1H), 4.42 (s, 1H), 4.38–4.19 (m, 6H), 4.18–4.04 (m, 2H), 3.78–3.71 (m, 1H), 3.25–3.09 (m, 4H), 2.71–2.60 (m, 2H), 2.54 (s, 4H). ^13^C NMR (101 MHz, DMSO) δ 159.08, 156.64, 153.11, 150.09, 149.60, 140.59, 138.28, 121.84, 120.07, 119.97, 84.50, 84.18, 79.65, 79.60, 77.93, 77.88, 66.21, 66.16, 65.88, 65.83, 29.55, 29.48, 29.43, 29.29. ^31^P NMR (162 MHz, DMSO) δ −1.37, −4.17. MALDI-TOF-HRMS: calcd for C_34_H_38_N_12_NaO_10_P_2_S_4_ [M + Na]^+^ 987.1090, found: 987.1088.

### 3.2. Biological Evaluation

Uptake of DTE-dCDN Prodrug. THP-1 cells were seeded at a concentration of 5 × 10^6^ cells/mL into a 12-well plate. After 16 h of incubation at 37 °C in a 5% CO_2_ atmosphere, the compounds were added to cell cultures and the cells were collected at time points of 1 h, 2 h, 4 h, 6 h, 8 h, 12 h, and 24 h by centrifugation for 1 min at 1000× *g*. The media was removed completely, and the cells were washed twice with 800 μL of ice-cold PBS before being lysed with 400 μL ice-cold 70% methanol. Lysates were incubated overnight at −20 °C and then centrifuged for 20 min at 13,000× *g* to remove cellular debris. The supernatant was analyzed via LC/MS/MS (Agilent 1290 Infinity II/6470A)

Prodrug **9** release parent drug initiated by cell lysates, DTT, and GSH. HEK 293T cells were plated in a 6-well plate at a density of 10^6^ cell per well. After 48 h, the culture media was removed and washed with 500 μL PBS buffer. Cells were collected into 1.5 mL microtubes. An amount of 200 μL weak RIPA lysis buffer containing 1 mM protease inhibitor PMSF (LEAGENE, Trenton, NJ, USA) was added to each tube for 30 min. After centrifugation at 13,000 rpm for 10 min, the supernatant (180 μL) was transferred into a new tube. An amount of 100 μM prodrug **9** was added into the cell lysate (60 μL) in an air-bath incubator at 37 °C. Aliquots of the reaction mixture were collected at various time points (0.5 h and 2 h) and stopped with 50 μL MeCN and 50 μL H_2_O followed by centrifugation at 13,000 rpm for 5 min. Finally, 10 μL of each aliquot was injected directly into the HPLC (Agilent 1260 equipped with a UV detector; column Agilent ZORABX SB-C18 5 μm [4.6 × 150 mm]; detection at 254 nm; column temperature 25 °C) for analysis. Amounts of 10 mM TEAA buffer (solvent A) and MeCN (solvent B) were used as the mobile phase with a flow rate of 1 mL/min. The gradient was set as the following: 0–2 min: 98% A/2% B; 2–6 min: 98% A/2% B to 100% B; 6–10 min: 100% B; 10–13 min: 100% B to 98% A/2% B; 13–15 min: 98% A/2% B.

Amounts of 10 mM DTT or GSH were added into prodrug **9** (100 μM) in an air-bath incubator at 37 °C for 0.5 h. Afterwards, 10 μL of each mixture was injected into HPLC for analysis. The HPLC methodology was the same as above.

In vitro serum stability assay. Each compound (100 μM) was incubated at 37 °C in the reaction buffer including 20% fetal bovine serum (FBS, GIBCO, Thermo Fisher Scientific, Waltham, MA, USA), 10 mM PBS (pH 7.4), and 1 mM MgCl_2_. At various times, aliquots of the reaction mixture were collected and stopped by adding 50 μL MeCN and diluted with 50 μL water. The reaction mixture was then centrifuged at 10,000 rpm for 5 min, leaving 100 μL supernatant for the HPLC assay. An amount of 10 μL of each aliquot was injected directly into HPLC (Agilent 1260 Infinity HPLC equipped with a UV detector; column Agilent ZORABX SB-C18 5 μm [4.6 × 150 mm; detection at 254 nm; column temperature 25 °C] for analysis. Amounts of 10 mM TEAA buffer (solvent A) and MeCN (solvent B) were used as the mobile phase with a flow rate of 1 mL/min. The gradient was set as the following: 0–2 min: 98% A/2% B; 2–6 min: 98% A/2% B to 100% B; 6–10 min: 100% B; 10–13 min: 100% B to 98% A/2% B; 13–15 min: 98% A/2% B. The % remaining of the test compounds after incubation in the serum was then calculated.

THP1-Lucia cell-based reporter assay. THP1-Lucia cells were seeded into a 24-well plate at a density of 5 × 10^4^ cells/well. After 16 h of incubation at 37 °C in a 5% CO_2_ atmosphere, serially diluted compounds were added to cell cultures and incubated for 24 h at 37 °C in a 5% CO_2_ atmosphere. Finally, the medium was collected to determine the luciferase activity using QUANTI-Luc (InvivoGen, San Diego, CA, USA) according to the manufacturer′s instruction to calculate the EC_50_ values of the tested compounds. The results are expressed as the mean ± SD from three independent experiments. Emax values were normalized to the response induced by 2′,3′-cGAMP in each assay and expressed as a mean percentage ± SD from three independent experiments. The THP1-Lucia cell line was obtained from Prof. Junmin Quan from Peking University (Beijing, China) [21].

Real-time quantitative PCR (RT-qPCR). The mRNA expression levels of diverse cytokines were measured using quantitative RT-PCR (RT-qPCR) assays. THP-1 cells or CT26 cells were incubated with the compounds for 4 h, subsequently harvested for RNA isolation according to the manufacturer’s instructions of RNApure Tissue&Cell Kit (DNase I) (CWBIO, Cambridge, MA, USA catalogue no. CW0560S). Next, 1 μg RNA was reversely transcribed to cDNA by TransScript All-in-One First-Strand cDNA Synthesis SuperMix for qPCR (One-Step gDNA Removal) (TransGen, Beijing, China). To examine the mRNA levels, real-time PCR was performed in a CFX96 Real-Time PCR system (Bio-RAD, Hercules, CA, USA) using PerfectStart Green qPCR Super Mix (TransGen) and primers (Sangon Biotech, Shanghai, China). The RT-qPCR primers used in this study are listed in Appendix A. The expression level was calculated according to the formula (2^−∆∆Ct^) using the GAPDH or β-actin as the internal reference gene and the relative gene expression fold was normalized to the DMSO-treated cells. The THP-1 and CT26 cell lines were obtained from American Type Culture Collection (ATCC, Manassas, VA, USA).

Enzyme-Linked Immunosorbent Assay (ELISA). THP-1 cells were incubated with the compounds for 4 h. Cell culture supernatants were then analyzed via ELISA for human IFN-β levels (ABclonal, Woburn, MA, USA, catalogue no. RK01630) according to the manufacturer’s instructions.

Cell viability of THP-1 and HEK293T cells. A total of 5 × 10^3^ HEK293T cells or THP-1 cells were seeded in 96-well plates. After 16 h of incubation at 37 °C in a 5% CO_2_ atmosphere, the cells were stimulated with indicated concentrations of the compounds (10 μM) for 24 h. Cell counting kit-8 (CCK8) (US EVERBRIGHT, Newport Beach, CA, USA, c6005) was then used to quantitatively assess the cell viability through the OD value at 450 nm. The HEK293T cell line was obtained from American Type Culture Collection (ATCC).

Western blotting. THP-1 cells (8 × 10^5^ cells/well) were seeded into a 6-well plate and treated with prodrug **9**, 2′,3′-cGAMP, 3′,3′-c-di-dAMP, and ADU-S100 for 4 h. The total proteins from the cells were extracted via cold RIPA buffer supplemented with protease inhibitors. Cell lysates were harvested via 1 × SDS-PAGE sample loading buffer (Cat. code: P0015, Beyotime, Shanghai, China), boiled at 100 °C for 10 min. Cell lysates were run on 10% SDS-PAGE gels and transferred onto NC membranes. The membranes were blocked with 5% BSA for 1 h and incubated with primary antibodies against STING, phospho-STING (Ser366), TBK1, phospho-TBK1 (Ser172), IRF3, phospho-IRF3 (Ser386), or β-actin overnight at 4 °C. Horseradish peroxidase-conjugated secondary antibodies were used as secondary antibodies. Finally, the membrane was detected using enhanced chemiluminescence for imaging.

## 4. Conclusions

In summary, we combined the deoxyribose backbone and the reduction-responsive prodrug strategy to design and synthesize the DTE-dCDN prodrug **9**, which could efficiently enter into cells to release the parent dCDN and exhibit a higher bioactivity than its parent dCDN 3′,3′-c-di-dAMP, 2′,3′-cGAMP, and ADU-S100 in THP1-Lucia cells. Furthermore, prodrug **9** induced the highest expression of *IFN-β*, *CXCL10*, *IL-6,* and *TNF-α* mRNA through STING-dependent IRF and NF-κB pathway signaling in the THP-1 cell lines. Taken together, the DTE-dCDN prodrug showed a general approach to improve the cell penetration and cell activity of CDN, which could be a novel and potential STING agonist for the STING-based immunotherapy of cancer, infectious diseases, and other disorders.

## 5. Patents

Zhen Xi and Zhenghua Wang are coinventors on patent WO 2020/143740.

## Data Availability

Data are contained within the article and Appendix A.

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
