# Peer review of "Dithioethanol (DTE)-Conjugated Deoxyribose Cyclic Dinucleotide Prodrugs (DTE-dCDNs) as STING Agonist"

_ijms, 2023, doi:10.3390/ijms25010086_

Round 1

Reviewer 1 Report

Comments and Suggestions for Authors

I don't have any comments. This paper looks ok to me. 

Reviewer 2 Report

Comments and Suggestions for Authors

The manuscript entitled "Dithioethanol (DTE)-Conjugated Deoxyribose Cyclic Dinucleo-2 tide Prodrugs (DTE-dCDNs) as STING agonist" by Zhiqiang Xie et al. reports the design and synthesis of a STING agonist prodrug which possess a high cellular uptake efficacy and can be released by GSH. The synthesis has been confirmed by 1H-NMR. In vitro analysis demonstrated the prodrug 9 had fast cellular uptake rate, leading to a lower EC50. Furthermore, prodrug 9 could induce cytokines production.

Overall, the manuscript is well written. The study design is clear. However, the figure in the manuscript is not clear and lacks proper units. Additional cell lines need to be tested to further confirm efficacy as well as safety. In vivo efficacy study evaluating the antitumor effect as well as biodistribution is suggested to demonstrate the capability of prodrug 9 as a Target Cancer Therapy. The manuscript could be published after addressing the following questions.

Major Comments
1. Figure 2, the y-axis unit needs to be included in the figure. How much percentage of prodrug can be uptake into the cell. It is clear the uptake take of prodrug arrives peak before 4 hrs; however, if only a small amount of total drug has been uptake, the remaining free drugs might lead to safety concerns.

2. Line 105, author mentioned “intracellular levels of parent dCDN 3′,3′-c-di-dAMP peaked in 6h” the figure 2A didn’t include the parent molecules.

3. Study 2.2, “Fast uptake and efflux of DTE-dCDN prodrug” The current study mainly shown uptake of molecules, efflux was not investigated. If the author wants to include efflux profile into this study, analysis of medium also need to be conducted.

4.A serum stability test of prodrug 9 is needed. The drug release study has demonstrated the prodrug 9 can be released by GSH. To make it as a target delivery therapeutic molecule, it is equally important that the prodrug cannot be processed until it being uptake, thus the stability in blood circulation is important.

5.Figure 4, y-axis unit need to be clearly indicated in the figure.

6. Study 2.6, authors use THP-1 Monocyte as a model to demonstrate prodrug 9 could stimulate cytokines production. Accepting the similarity of THP-1 to monocytes, the result would be more comprehensive to test the molecules on PBMC or primary monocytes with a more complicated cluster.

7. HUVEC cells should also be tested for safety as well as cell uptake. The cellular uptake of prodrug 9 is not cell specific, after administration in vivo, it might be nonspecifically uptake to endothelial cells, leading to potential toxicity. Thus, HUVEC cell binding assay is suggested.

8. As a potential targeted cancer therapeutic molecule, in vivo efficacy and biodistribution study is highly recommended. The concentration of prodrug at tumor site needs to be high enough to achieve therapeutic effect; however, the prodrug could be nonspecifically bound to blood cells and/or healthy organs, leading to low efficacy and toxicity. These questions can only be answered by in vivo studies.

Comments on the Quality of English Language

Overall, the manuscript is well written. Few typo need to be corrected.

Reviewer 3 Report

Comments and Suggestions for Authors

This is an interesting manuscript addressing the development of new STING agonist molecule. While the data shown are promising, the authors must consider first the scientific rigor when a new drug from an existing class is developed. The question they must answer is: Is the new drug is more efficacious with less side effects than the STING agonists which are currently under investigation in clinical trials?  While the use of 2’,3’-cGAMP and 3’3’-c-di-dAMP as controls is acceptable, these are natural STING agonists with very short half-life and low cell permeability. The authors must include as control any STING agonist from biotech, which has already the structure changed to be more resistant to phosphodiesterase (e.g. ADU-S100) and also one with higher cell permeability (e.g. diABZI). These must be the real positive controls for their new drug.

Another big issue with this manuscript is that there is an incomplete investigation to claim that this new proposed agonist activates STING pathway. Therefore, the authors must include further data to prove the functionality of their drug, as follows:

·         Immunoblot data to clearly demonstrate STING activation, aka the phosphorylation of downstream molecules in STING pathway (P-STING, P-TBK1, p-IRF3).

·         Proteomic data (e.g., ELISA) to demonstrate that indeed, besides increasing the transcription of IFNb, the protein is translated and secreted outside of target cells. Transcriptome does not mean necessary the protein will be produced.

·         The authors must include at least one more cell line (preferable solid tumor origin) to validate their observed data in only ONE cell line. Since we do not know if STING agonist trigger innate immunity and shape anti-tumor effects via STING activation in tumor or in dendritic cells, the perfect case scenario will be to add the exploration of STING agonist in dendritic cells.

·         While this is a pre-clinical investigation, a mouse tumor model to show the anti-tumor effect of the new agonist will increase significantly the overall scientific value of the author’s work.

As a separate note the experiment using cGAS inhibitor is not relevant. cGAS is an upstream molecule in STING pathway and STING expression/activation does not positively or negatively regulate cGAS.

Round 2

Reviewer 2 Report

Comments and Suggestions for Authors

I appreciate author's supplementary data and responses. The manuscript has been improved and most of my questions has been addressed. The manuscript can be considered for publication in IJMS. 

Reviewer 3 Report

Comments and Suggestions for Authors

I would like to congratulate the authors for their extensive work to answer to all reviewer's questions. Therefore, I think that the manuscript suitable for publication in the present form.

Thank you.